# Microsurgical Strategies in Post-Radiation and Revision Breast Reconstruction: Optimizing Outcomes in High-Risk Patients

**DOI:** 10.3390/cancers17233831

**Published:** 2025-11-29

**Authors:** Thomas J. Sorenson, Carter J. Boyd, Oriana Cohen, Mihye Choi, Nolan Karp

**Affiliations:** Hansjorg Wyss Department of Plastic Surgery, NYU-Langone Health, New York, NY 10016, USA; thomas.sorenson@nyulangone.org (T.J.S.);

**Keywords:** microsurgery, breast, revision, reconstruction, implant based breast reconstruction, radiation

## Abstract

Patients who need breast reconstruction after radiation treatment or after a previous reconstruction has failed often face more difficult surgeries. Radiation and earlier operations can damage blood vessels, thin the skin, and create scar tissue, which makes new reconstruction more complex and increases the chance of complications. This review explains the surgical approaches that can help doctors achieve safer and more reliable results for these high-risk patients. We describe how surgeons choose the best tissue from different parts of the body, how they adjust their techniques to work around damaged areas, and how imaging tools that show real-time blood flow can guide safer decisions during surgery. We also outline strategies for patients who need a revision of a prior implant-based reconstruction, including removing scarred tissue, replacing missing skin, and using healthy tissue to rebuild the breast. Finally, we discuss the importance of careful planning, selection of healthy blood vessels, and close monitoring after surgery. By understanding the challenges and available solutions, this work may help improve outcomes, reduce complications, and support better long-term well-being for people who need reconstruction in the setting of radiation or prior surgical failure.

## 1. Introduction

Breast reconstruction has become an integral component of comprehensive breast cancer care with autologous microsurgical techniques offering superior long-term aesthetic and functional outcomes compared to implant-based reconstruction in many patients [1]. With the passage of the Women’s Health and Cancer Rights Act of 1998 and the rise in breast conserving therapy, a growing group of patients present with breast reconstructive needs after prior radiation therapy, previous failed reconstruction, or both [2,3,4]. These patients represent a relatively high-risk cohort where the reconstructive landscape may be altered by compromised vascularity, scarred recipient sites, tissue fibrosis, and distorted anatomy resulting in elevated complication rates [5,6,7,8].

In these radiated and revision scenarios, microsurgical reconstruction may be the most definitive and durable option [9,10]. Advances in perforator flap techniques, alternative donor site utilization, and recipient vessel planning have expanded the reconstructive toolbox available to surgeons managing these high-risk cases. Further, intraoperative imaging technologies such as indocyanine green (ICG) angiography, and the widespread adoption of enhanced recovery protocols have additionally contributed to improved outcomes in this population [11,12,13]. Nevertheless, the challenge of these patients demands careful preoperative planning, surgical adaptability, and diligent postoperative management. This narrative review that draws on published literature and our personal experiences aims to provide a comprehensive overview of current microsurgical strategies for breast reconstruction in post-radiation and revision settings and offers a practical framework for breast reconstructive surgeons facing these challenging clinical scenarios.

## 2. Patient Evaluation and Risk Stratification

Successful microsurgical breast reconstruction in the post-radiation or revision setting begins with rigorous patient evaluation and individualized risk stratification. Particular attention should be paid to radiation, any prior reconstruction attempts, and comorbid conditions. Noninvasive imaging also plays an important role in preoperative evaluation and operative planning with computed tomography angiography (CTA) being the gold standard for perforator mapping and donor site assessment [14]. Some authors also advocate for the use of color duplex ultrasound (CDU) for preoperative flap planning and design, though we do not perform this [15].

Patient risk stratification can be broadly based on systemic and local factors. Lese et al. developed a novel predictability index with three classes for systemic factors that included coronary heart disease, diabetes, smoking, peripheral artery vascular disease, arterial hypertension and found patients with moderate-risk index had almost 10 times higher chances of developing vascular compromise than those in the low-risk group, while a high-risk index had almost 20 times higher odds [16]. High-risk local factors include prior chest wall radiation, previous flap or axillary dissection, history of infected or exposed implants, severe capsular contracture, or thin mastectomy skin flaps [17]. A thorough risk assessment should guide decision-making, and patients with multiple risk factors may benefit from a more staged approach [16]. Additionally, any new or chronic wounds noted on evaluation, especially in the setting of prior radiation, must be formally analyzed by surgical pathology for possible breast cancer recurrence.

The optimal timing of autologous breast reconstruction following radiation therapy remains a subject of ongoing debate. Some studies support immediate reconstruction in select patients, citing acceptable complication rates [18,19]. A 2021 meta-analysis by Heiman et al. found immediate free flap breast reconstruction associated with superior flap survival compared with delayed reconstruction with comparable complication rates [5]. However, delayed reconstruction, typically performed 6 to 12 months after completion of radiation, has traditionally been favored by our group to allow for resolution of acute inflammation, tissue fibrosis stabilization, and revascularization of the chest wall. Importantly, no standardized interval has been universally adopted, and current data remain limited by heterogeneity in study design and patient selection. As such, timing should be individualized, incorporating factors such as radiation modality, patient comorbidities, reconstructive goals, and the surgeon’s experience with high-risk flaps.

## 3. Flap Selection and Design

In patients with prior radiation or previously failed reconstruction, abdominal-based flaps (e.g., deep inferior epigastric perforator [DIEP] flap) remain foundational, but surgeons must be prepared to employ alternative donor sites (e.g., thigh, back), utilized stacked or bipedicled flaps, or customize flap design to optimize perfusion, volume and inset into a compromised recipient site [20] (Figure 1).

### 3.1. Abdominal-Based Flaps: The Gold Standard

Even in radiated or revision breasts, the DIEP flap remains the workhorse for autologous reconstruction due to its reliable anatomy, low donor site morbidity, and favorable aesthetic contour. However, in thin patients with larger defects, volume requirements may exceed what a unilateral DIEP can offer. In these scenarios, we like to consider stacked DIEP flaps, which can augment volume without breast implants with lower risk of fat necrosis compared to non-stacked flaps [21,22,23]. For patients with extensive irradiated skin or unreliably thin skin envelope, a bipedicled DIEP flap or muscle-sparing transverse rectus abdominus myocutaneous (MS-TRAM) flap can be helpful at providing perfusion across a wider skin paddle though we find that sufficient skin can usually be obtained with a standard unipedicled DIEP alone [24]. However, patients with midline scars, prior abdominal surgeries or inadequate tissue volume may not be ideal abdominal-based flap candidates, indicating the need for secondary donor sites.

### 3.2. Alternative Donor Sites

When necessary, several alternative donor sites have emerged as valuable alternatives. We prefer the profunda artery perforator (PAP) flap, because the ease of pedicle dissection, the pliable tissue for irradiated fields with constricted breast envelopes, ability to be stacked or used bilaterally, and no position change [25]. Systematic review by Jo et al. comparing PAP flap to transverse upper gracilis (TUG) flap found comparable flap loss with decreased donor site morbidity and increased flap volume when using the PAP flap [26]. Due to its advantages, some authors [27] even recommend that the PAP flap be considered a primary therapeutic option and not as an alternative to the DIEP flap though we do not yet view it that way.

Posteriorly based flaps are also options though they generally have a highly technical pedicle dissection and require a patient position change between flap harvest and inset making them generally less ideal [28]. The lumbar artery perforator (LAP) flap is a valuable alternative option and is a high density fat flap that is less pliable but ideal for volume replacement [29]. Numerous series report favorable outcomes with reported complications related to seroma and scarring and 97% of patients recommending the surgery to someone in similar position [30,31]. Few comparisons exist between posterior flap options (LAP, superior gluteal artery perforator [SGAP]) though a CT-based anatomical study found LAP flaps had thicker subcutaneous fat and a larger vascular pedicle diameter, whereas the SGAP flaps had a longer vascular pedicle. If needed, these alternative donor sites can be combined with abdominal donor sites successfully [32,33].

### 3.3. Modifying Flaps for Irradiated Fields

Radiated mastectomy beds present particularly hostile environments for flap integration. In patients with thin mastectomy skin, we like to place deepithelialized flaps underneath the mastectomy skin flaps to reinforce the skin with vascularized tissue. Further, we will also externalize a skin island to expand the skin envelope [34]. Volume overcorrection is also a useful strategy to account for flap resorption in irradiated beds, though care must be taken to prevent fat necrosis from overextending the vascular pedicle [23]. Lastly, utilize geometric principles and inset flaps in conical shape to restore maximum projection which can be useful in contracted breast pockets [25]. In certain circumstances with substantial contraction after radiation, we find breast capsulectomy helpful to soften and expand the breast pocket though it should only be performed judiciously if there is concern for mastectomy skin flap viability [35]. In these severely contracted capsules, radial scoring is useful if capsulectomy is not possible.

### 3.4. Hybrid and Fat-Augmented Flaps

Lastly, in cases where complete autologous reconstruction is not feasible, hybrid reconstruction with flap plus implant or flap plus fat grafting may be viable options [36]. In patients desiring breast volume without a large volume flap, implants can be used successfully under smaller autologous tissue, which provides healthy, vascularized coverage over the implant to reduce capsular contracture and overall complication risk in irradiated fields [6,37]. Finally, serial fat grafting with has been shown to improve contour and quality of overlying irradiated skin with patients experiencing improved skin quality and reduced capsular contracture at six months [38,39,40,41]. However, it is important to note that there is an increased rate of fat necrosis and infection in irradiated breasts as compared to non-irradiated ones [42,43]. There are numerous protocols, but most recommend repeated fat implantation no fewer than every 20 days until obtaining a stable result and the patient is satisfied [42].

In select patients, exclusive autologous fat grafting (lipofilling-only breast reconstruction) has emerged as a less invasive alternative to flap or implant-based methods. While lipofilling can improve skin quality and modestly restore volume, especially in radiated fields, multiple sessions (often three to five) are typically required to achieve acceptable aesthetic outcomes [44]. Furthermore, in high-risk patients with substantial soft tissue deficits or poor skin envelope quality, lipofilling alone may be insufficient to achieve durable breast mound reconstruction [45]. Nonetheless, for patients who decline more invasive procedures or have limited donor site availability, lipofilling remains a valuable adjunct or alternative [46].

## 4. Intraoperative Techniques and Adjuncts

In the setting of post-radiation and revision breast reconstruction, the margin for error is slim, and novel intraoperative technologies have become critical tools for microsurgeons seeking to minimize variables. One of the most impactful advances in microsurgical decision-making is real-time perfusion imaging, particularly with indocyanine green (ICG) angiography, which has been found to reduce rates of severe mastectomy flap necrosis, and reoperation [47,48]. However, these data are not consistent with other authors finding no significant differences in mild flap necrosis, nipple necrosis, fat necrosis, implant exposure, flap loss, or the number of overall complications [49,50]. Ultimately, a Cochrane review found that no high-quality conclusions can be drawn about what method of skin flap assessment is best [51].

When performing immediate reconstruction in radiated settings, we especially use this tool to assess for skin flap vascularity after mastectomy before flap inset and identify hypoperfused zones for excision [11]. If we anticipate extensive skin flap loss due to hypoperfusion, we will inset the fully epithelialized flap with epidermis buried in anticipation of future use after the mastectomy skin demarcates and requires debridement. We do not routinely use ICG when performing delayed reconstruction in radiated or revision settings unless there is concern after managing the capsule and preparing the breast pocket for the flap. Other emerging tools include hyperspectral imaging (HSI), which is noninvasive and dye-free and provides tissue oxygenation maps, and laser-assisted fluorescence angiography (LAFA) which can provide quantitative flow metrics, though definitive data are still unsettled [52]. We do not routinely use either of these technologies.

Flap inset in high-risk patients also requires strategic forethought. When present, skin paddle orientation should anticipate scar lines, prior incisions, and areas of skin deficiency. Importantly, in cases of previous implant-based reconstruction, excision of any residual breast capsule should be performed as necessary to ensure tension-free inset and prevent area for fluid accumulation. This also releases areas of constriction and allows the breast pocket to expand, which helps to improve final breast shape. Additionally, the use of quilting sutures to laterally control the pocket and judicious use of closed-suction drains minimize seroma formation, particularly in revision or radiated patients where tissue planes are disrupted.

## 5. Recipient Vessel Strategies in the Vessel-Depleted Chest

In microsurgical breast reconstruction, the choice of recipient vessels is pivotal in high-risk patients. Prior radiation, axillary dissection, central venous access, or failed ABR attempts may compromise first line recipient vessels and make them inaccessible. These “vessel-depleted” scenarios require flexibility, familiarity with alternative options, and creative intraoperative problem-solving. These options are summarized in Table 1.

### 5.1. The Internal Mammary System: First Line but Not Infallible

The internal mammary artery and vein (IMA/V) remain the preferred recipient vessels for most microsurgical breast reconstructions due to their consistent anatomy, location, and caliber [53]. However, in irradiated or previously operated fields, these vessels are highly prone to insult or injury due to their central location in the chest and can be calcified or fibrotic, complicating the anastomosis. Strategies for successfully using these vessels include performing end-to-end anastomoses, avoiding artery clamps on calcified vessels, and using a limited amount of double-armed microsutures with inside-outside directed bites which can prevent dislodgement of calcific plaques. Some authors even advocate for use of fibrin sealants instead of “rescue” sutures to minimize intravascular trauma though we do not routinely perform this [54]. In patients with prior failed ABR, prior costal cartilage resection may prevent preferred access (third intercostal space) to the IMA/V for anastomosis. In these situations, we will perform anastomosis proximally in the second intercostal space or distally using retrograde flow before looking to alternative sites.

### 5.2. The Thoracodorsal System: Secondary but Valuable

The thoracodorsal artery and vein (TDA/V) is a valuable second choice but may be unusable in patients with prior axillary lymph node dissection, prior *Latissimus dorsi* (LD) muscle flap harvest, or radiation-induced scarring in the axilla [55,56]. However, when viable, the TDA/V can still be a valuable option in hybrid reconstruction based around the LD muscle. Teotia et al. described success in 21 breasts using the thoracodorsal and serratus vessels for salvage situations [57]. However, the overall reduced pedicle length generally restricts the location of the flap tissue to a more lateral position on the chest.

### 5.3. Salvage Options in the Vessel-Depleted Chest

If the IMA/V or TDA/V are unusable, there are several alternatives to consider. The thoracoacromial vessels are often available in the clavipectoral triangle and can be accessed via an infraclavicular incision or through the breast pocket [58,59]. In situations of extreme salvage, the contralateral IMA/Vs may be a viable option [60]. Similarly, the intercostal vessels are rarely used but can be valuable in cases of extreme salvage [61]. For salvage venous outflow, we will use the cephalic vein, which is superficial, easily accessible in the deltopectoral groove and very reliable for providing outflow in vein graft scenarios.

### 5.4. Vein Grafts and Arteriovenous (AV) Loops

When recipient vessels are too short, scarred, or absent, interpositional vein grafts or AV loops may be required [62,63]. There are numerous sources throughout the body. The saphenous vein is long, of appropriate caliber, and easily harvestable from the leg to use as an interpositional vein graft [64]. The cephalic vein can be used as a free interpositional vein graft or turned up and used for short bridging situations (reverse flow) [65,66]. Lastly, in cases of extreme salvage, the saphenous vein can be used as a one-stage or two-stage AV loop with the axillary or subclavian artery/vein to allow sufficient pedicle length to tunnel into the flap site. Especially in irradiated or damaged fields, vein grafts and AV loops are associated with increased complications but can remain crucial salvage tools for capable surgeons [67].

### 5.5. Intraoperative Decision-Making and Contingency Planning

In these patients, preoperative evaluation must include vessel imaging to maximize preoperative planning and minimize intraoperative surprises. Preoperative imaging (CTA or MRA) should always include recipient vessel mapping and a close working relationship with the radiologist is critical initially to obtain valuable imaging studies in the correct phase. Lastly, during these challenging cases, surgeons must be prepared to explore multiple intercostal spaces, perform extensive pedicle dissection, convert to an alternative recipient vessel plan intraoperatively, including the use of vein grafts or AV loops when pedicle reach is marginal.

## 6. Revision-Specific Strategies

Revision breast reconstruction is among the most technically and psychologically complex domains in reconstructive surgery. Patients may present requiring prior IBBR explantation, after ABR flap loss, or with capsular contracture or another unsatisfactory cosmetic outcome. These situations may also be layered atop a background of radiation, further complicating reconstruction. These cases demand a tailored, multidisciplinary approach that prioritizes safety, aesthetic restoration, and reasonable patient-centered goals.

### 6.1. Management of IBBR Explantation and Re-Reconstruction

A patient with IBBR requiring explantation due to seroma or implant exposure represents a high-stakes challenge as these patients frequently end up with extensive scarring and soft tissue loss. After removing the implant, numerous strategies exist to attempt to preserve the breast pocket and reconstruction, including use of continuous pocket irrigation and intravenous (IV) antibiotics. In a comparative study, Haque et al. reported that patients treated with continuous pocket irrigation had higher satisfaction, received a new implant earlier, and had fewer admissions and outpatient visits than those with delayed reinsertion but incurred higher average costs, longer inpatient stays, and underwent more procedures [68]. These largely equivocal results are supported by a 2025 systematic review and other studies [69,70,71]. We do not routinely do this in our practice.

Our strategy is to remove the implant, treat the infection and perform delayed autologous reconstruction after interval healing [72]. In those patients with highly scarred breast pockets and/or fibrotic skin due to infection or radiation effects, we do not use tissue expansion prior to autologous reconstruction and instead use more skin from the donor to create a breast envelope. If desired, additional stages can involve fat grafting, or symmetrizing procedures for final aesthetic optimization. Ultimately, these patients typically have a prolonged clinical course and must be counseled as such to minimize surprises and reconstruction fatigue, and those patients with periprosthetic infection have greater than 10 times odds for conversion to autologous reconstruction [73].

### 6.2. Management of Unsatisfactory Cosmetic Outcomes

Capsular contracture remains one of the most frequent reasons for revision in IBBR, particularly in the setting of radiation. These scenarios require a multimodal treatment approach, including complete capsulectomy, especially in the setting of Baker grade III/IV contracture or suspected biofilm, and conversion to autologous reconstruction to provide vascularized tissue for pocket expansion [73,74,75] (Figure 2). Finally, biologic or bioresorbable mesh (e.g., P4HB, ADM) can be used successfully in these well-vascularized beds to stabilize implant position within the neo-pocket [76,77].

## 7. Postoperative Management and Recovery

Postoperative care plays a pivotal role in determining the long-term success of microsurgical breast reconstruction as even technically flawless reconstructions are vulnerable to failure if not supported by structured postoperative protocols. For enhanced recovery after surgery (ERAS), we attempt to optimize patients preoperatively as much as possible though we do not find the need to attempt specific nutritional or behavioral optimization [78]. As part of our protocol, we use regional blocks intraoperatively which we find improves early ambulation and does minimize the use of opioid analgesics [79]. We typically use liposomal bupivacaine though there is no strong evidence to support the use of this over standard bupivacaine [80]. We do not routinely use flap monitoring adjuncts, like implantable Dopplers or near-infrared spectroscopy, and find that monitoring these flaps, even in the setting of radiation skin changes, can be performed successfully with standard serial clinical examinations.

Routine breast flap patients are frequently discharged on post-operative day (POD) two or three depending on the procedure, but these complex or revision patients are frequently kept in the hospital longer for additional monitoring. Upon discharge, clear, structured instructions are especially important in these patients, including signs of flap compromise, drain care and infection prevention, activity restrictions, and avenues for psychosocial support during the healing process. We especially like to use cancer-specific psychosocial support services, which we find help patients navigate the unique challenges posed by their disease [81]. Empowering patients with education and frequent follow-up improves early detection of issues and promotes engagement in long-term care and optimal outcomes.

## 8. Outcomes

Assessing outcomes in high-risk breast reconstruction patients requires a nuanced interpretation of the available evidence. While microsurgical reconstruction offers durable, aesthetically favorable results in experienced hands, the literature reflects significant variability in complication rates, success metrics, and patient-reported outcomes in this subgroup.

### 8.1. Flap Survival and Complication Rates

Autologous free flap reconstruction continues to demonstrate high overall success rates (>95%) even in high-risk populations [20]. These rates remain high even in radiated breasts with flap loss rates less than 5% [82,83]. However, radiation-associated cases have been found to have substantially increased risk of wound healing complications, higher rates of fat necrosis, and increased rates of unplanned return to the OR, particularly within the first 30 days [5]. Recommendations generally favor autologous reconstruction in a delayed fashion after radiation therapy [84,85]. Revision cases, especially those involving prior implant removal, infected fields, or multiple surgeries, tend to have higher technical difficulty and ischemia times, higher use of vein grafts or salvage vessel strategies, and longer hospital stays and increased drain durations [86,87].

### 8.2. Aesthetic and Functional Outcomes

Several studies using validated patient-reported outcome measures (PROMs), such as the BREAST-Q, have found higher satisfaction with autologous reconstruction, especially in irradiated patients, and substantially improved psychosocial and physical well-being scores after successful revision [88,89,90]. Mean BREAST-Q scores are typically 6–10 points higher than IBBR at two years post-op, and this difference persists up to 10 years out [91,92,93]. Further, patients who undergo salvage autologous reconstruction also report quality-of-life improvements (though less than had they undergone immediate or delayed autologous reconstruction initially), highlighting the restorative potential of well-executed microsurgical revision [94].

### 8.3. Evidence Gaps and Limitations

Despite a growing body of literature, evidence specific to post-radiation and revision patients remains limited as most large studies combine primary and revision cases, obscuring differential risks. Furthermore, radiation regimens are heterogeneous, making comparison difficult, and few prospective studies stratify outcomes by vessel depletion or prior surgical history. There is a need for prospective, risk-adjusted outcome registries and standardized reporting of complications, particularly in flap salvage scenarios and staged reconstructions.

## 9. Conclusions

Microsurgical reconstruction in patients with prior radiation or failed reconstruction is technically feasible and increasingly common but carries elevated risks and requires thoughtful adaptations of technique. These high-risk patients often arrive with depleted recipient vessels, compromised skin envelopes, and a history of surgical disappointment. While various flaps and recipient vessels have been successfully employed, current evidence does not definitively establish the superiority of one approach, or the use of adjuncts, over another in these settings. Future research should prioritize prospective multicenter data collection, evaluation of long-term outcomes, cost-effectiveness, and the development of validated decision-support tools to guide reconstruction in this high-risk population.

## Figures and Tables

**Figure 1 cancers-17-03831-f001:**
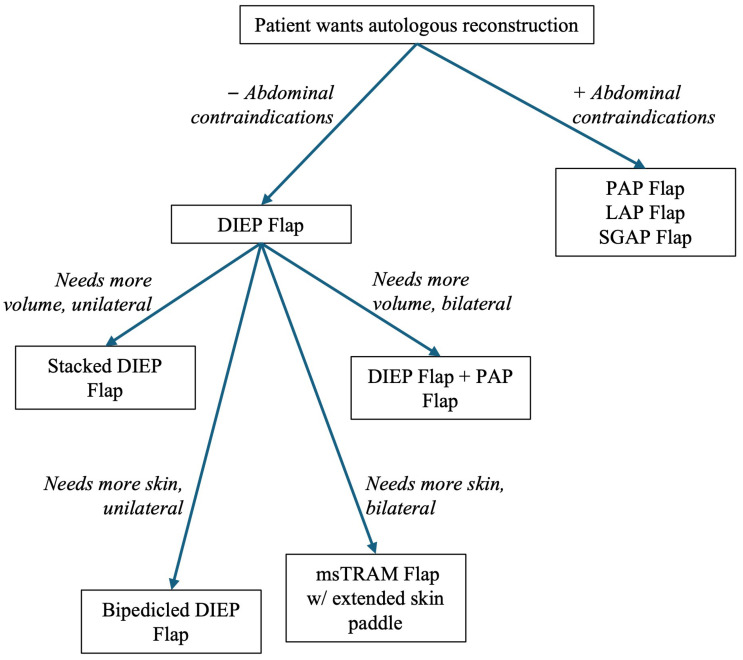
Framework for selecting the appropriate autologous reconstruction method.

**Figure 2 cancers-17-03831-f002:**
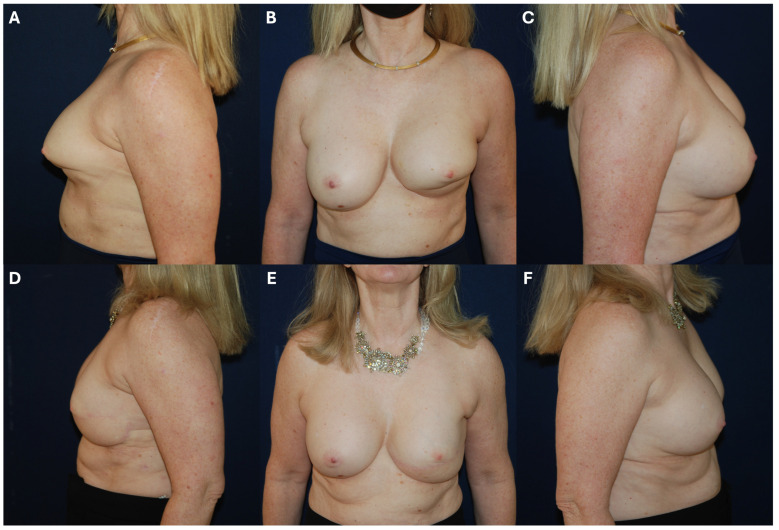
62-year-old female patient with remote history of bilateral breast augmentation and recent history of left breast lumpectomy in 2020 followed by radiation therapy presents with left inferior breast radiation contracture (**A**–**C**) requiring left pedicled *Latissimus dorsi* flap reconstruction with implant exchange and capsulectomy (**D**–**F**).

**Table 1 cancers-17-03831-t001:** Recipient Vessel Options in Post-Radiation and Revision Breast Reconstruction.

Recipient Vessel	Advantages	Disadvantages	Ideal Scenarios
Internal Mammary Vessels (IMV)	- Consistent anatomy - Medial flap inset - Typically first choice	- Often compromised by prior radiation - May require rib/cartilage sacrifice	- Non-radiated chest - First-time recon - Favorable imaging (CT angiography)
Thoracodorsal Vessels (TDV)	- Preserved in many mastectomies - Reliable location	- May be damaged in prior axillary dissection - More lateral inset	- Prior radiation to chest - Salvage after IMV unusable
Circumflex Scapular Vessels	- Rarely used in prior surgery - Often preserved	- Technically demanding - Short pedicle - Lateral inset	- Salvage after IMV and TDV failure - Local flap options needed
Cephalic Vein (for venous outflow)	- Accessible superficial vein - Can be used as additional outflow	- Small caliber - Not arterial - Risk of venous hypertension	- As adjunct for venous congestion in cases of poor flow
Transverse Cervical Vessels	- Outside prior radiation zones - Long pedicle match	- Requires supraclavicular dissection - Position limits flap inset options	- Extensive prior chest radiation - Total vessel loss in chest and axilla
Contralateral IMV (Cross-chest)	- Avoids radiated field - Straightforward anatomy	- Longer flap reach needed - Risk of contour asymmetry - Technically complex	- Unilateral flap with unilateral chest radiation - No ipsilateral options
AV Loop Graft (saphenous vein graft to axillary or subclavian)	- Bypasses all compromised native vessels - Useful for complete vessel loss	- Two-stage - Thrombosis risk - Technically demanding	- Multiple failed reconstructions - Prior radiation and vessel depletion

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
