# Peer review of "Microsurgical Strategies in Post-Radiation and Revision Breast Reconstruction: Optimizing Outcomes in High-Risk Patients"

_cancers, 2025, doi:10.3390/cancers17233831_

Round 1

Reviewer 1 Report

Comments and Suggestions for Authors

The review is well conducted and didactic. I suggest to improve the clinical and didactic message adding some figures about the proposed flaps, and some case report could be usefull to better understand the surgical procedure and the outcome to non plastic surgeon colleagues.

Author Response

Manuscript Number: cancers-3895389

Title: Microsurgical Strategies in Post-Radiation and Revision Breast Reconstruction: Optimizing Outcomes in High-Risk Patients

Reviewer #1:

  1. The review is well conducted and didactic. I suggest improving the clinical and didactic message adding some figures about the proposed flaps, and some case report could be useful to better understand the surgical procedure and the outcome to non-plastic surgeon colleagues.

We sincerely thank the reviewer for the positive feedback and thoughtful suggestions. In response, we have strengthened the clinical and educational impact of the manuscript by adding an illustrative figure that demonstrate key autologous flap strategies used in post-radiation and revision breast reconstruction to contextualize these outcomes in a real-world clinical scenario (Figure 2).

Reviewer 2 Report

Comments and Suggestions for Authors

Dear Authors we read with interest your review that addresses microsurgical breast reconstruction in high-risk patients, specifically those with prior radiation therapy or failed reconstruction. It synthesizes current literature and expert strategies, including flap selection, recipient vessel approaches, intraoperative technologies, and revision-specific considerations. The topic is timely and clinically relevant, offering practical insights for reconstructive surgeons.

We think the review is comprehensive but reads at times like a surgical manual rather than a critical synthesis. To improve impact, the authors should better distinguish their novel contributions (frameworks, algorithms, expert consensus points) from what is already well-established.

In addition to that the manuscript is long and dense. While thorough, sections could benefit from greater synthesis and less repetition. For example, outcomes, limitations, and psychological recovery could be consolidated for clarity.

Moreover figures or summary tables could enhance readability (e.g., a decision algorithm for flap selection in irradiated/revision cases, or a table of recipient vessel options with pros/cons).

The review does not clearly state the methodology for literature selection. Was this a narrative review or a structured/scoping review? Even if not systematic, some explanation of how references were chosen would strengthen transparency.

Where conflicting evidence exists (e.g., timing of autologous reconstruction post-radiation, efficacy of ICG angiography), this should be explicitly acknowledged.

We think Figure 1 is useful and we suggest to consider adding more visual aids.

 Well-cited and up-to-date, please add The Effect of Adjuvant Radiotherapy on One- and Two-Stage Prosthetic Breast Reconstruction and on Autologous Reconstruction: A Multicenter Italian Study among 18 Senonetwork Breast Centres.

Emanuele Lisa AV, Salgarello M, Huscher A, Corsi F, Piovani D, Rubbino F, Andreoletti S, Papa G, Klinger F, Tinterri C, Testori A, Scorsetti M, Veronesi P, Leonardi MC, Rietjens M, Cortinovis U, Summo V, Rampino Cordaro E, Parodi PC, Persichetti P, Barone M, De Santis G, Murolo M, Riccio M, Aquinati A, Cavaliere F, Vaia N, Pagura G, Dalla Venezia E, Bassetto F, Vindigni V, Ciuffreda L, Bocchiotti MA, Sciarillo A, Renzi N, Meneghini G, Kraljic T, Loreti A, Fortunato L, Pino V, Vinci V, Klinger M.Breast J. 2023 May 9;2023:6688466. doi: 10.1155/2023/6688466.    The evolution of autologous breast reconstruction. Costanzo D, Klinger M, Lisa A, Maione L, Battistini A, Vinci V.Breast J. 2020 Nov;26(11):2223-2225. doi: 10.1111/tbj.14025. Epub 2020 Sep 10.PMID: 32909653 Review.

Author Response

Manuscript Number: cancers-3895389

Title: Microsurgical Strategies in Post-Radiation and Revision Breast Reconstruction: Optimizing Outcomes in High-Risk Patients

Reviewer #2:

  1. We read with interest your review that addresses microsurgical breast reconstruction in high-risk patients, specifically those with prior radiation therapy or failed reconstruction. It synthesizes current literature and expert strategies, including flap selection, recipient vessel approaches, intraoperative technologies, and revision-specific considerations. The topic is timely and clinically relevant, offering practical insights for reconstructive surgeons. We think the review is comprehensive but reads at times like a surgical manual rather than a critical synthesis. To improve impact, the authors should better distinguish their novel contributions (frameworks, algorithms, expert consensus points) from what is already well-established.

We thank the reviewers for their thoughtful and generous feedback. We appreciate the recognition of the review’s comprehensiveness and relevance, and we agree that its impact can be enhanced by more clearly distinguishing novel contributions from established practice. We have revised the manuscript to explicitly highlight our proposed decision-making frameworks and consensus strategies throughout, including clarifying where these differ from traditional approaches. We believe these revisions help elevate the manuscript to critical synthesis.

  1. In addition to that the manuscript is long and dense. While thorough, sections could benefit from greater synthesis and less repetition. For example, outcomes, limitations, and psychological recovery could be consolidated for clarity.

We thank the reviewers for this valuable feedback. We have revised our entire manuscript to improve brevity and synthesis, including consolidating sections that may be redundant and unnecessary.

  1. Moreover figures or summary tables could enhance readability (e.g., a decision algorithm for flap selection in irradiated/revision cases, or a table of recipient vessel options with pros/cons).

We thank the reviewers for this excellent suggestion. In response, we have added a comparative summary of recipient vessel options, detailing anatomic location, technical considerations, advantages, disadvantages, and preferred scenarios based on prior radiation or failed reconstruction (Table 1). We have also included additional figures of different flap selections to demonstrate our decision strategies. We believe these additions improve the manuscript’s didactic value and offer quick-reference tools for practicing surgeons and enhance the manuscript’s impact.

  1. The review does not clearly state the methodology for literature selection. Was this a narrative review or a structured/scoping review? Even if not systematic, some explanation of how references were chosen would strengthen transparency.

We thank the reviewer for this helpful comment. The manuscript was intended as a narrative review, focused on integrating current literature with institutional experience and expert opinion to guide microsurgical decision-making in complex cases. We have revised to manuscript to clarify (Lines 44-45).

  1. Where conflicting evidence exists (e.g., timing of autologous reconstruction post-radiation, efficacy of ICG angiography), this should be explicitly acknowledged.

We thank the reviewer for this excellent point. We agree that areas of conflicting evidence should be clearly acknowledged to present a balanced and transparent synthesis. We have now modified both areas to present a more balance assessment (Lines 117-119; Lines 65-73).

  1. We think Figure 1 is useful and we suggest considering adding more visual aids.

We appreciate this comment, and we agree. We have now included more visual aids to help relay our message clearly (Figure 2).

  1. Well-cited and up-to-date, please add

The Effect of Adjuvant Radiotherapy on One- and Two-Stage Prosthetic Breast Reconstruction and on Autologous Reconstruction: A Multicenter Italian Study among 18 Senonetwork Breast Centres. Emanuele Lisa AV, Salgarello M, Huscher A, Corsi F, Piovani D, Rubbino F, Andreoletti S, Papa G, Klinger F, Tinterri C, Testori A, Scorsetti M, Veronesi P, Leonardi MC, Rietjens M, Cortinovis U, Summo V, Rampino Cordaro E, Parodi PC, Persichetti P, Barone M, De Santis G, Murolo M, Riccio M, Aquinati A, Cavaliere F, Vaia N, Pagura G, Dalla Venezia E, Bassetto F, Vindigni V, Ciuffreda L, Bocchiotti MA, Sciarillo A, Renzi N, Meneghini G, Kraljic T, Loreti A, Fortunato L, Pino V, Vinci V, Klinger M.Breast J. 2023 May 9;2023:6688466. doi: 10.1155/2023/6688466.   

The evolution of autologous breast reconstruction. Costanzo D, Klinger M, Lisa A, Maione L, Battistini A, Vinci V.Breast J. 2020 Nov;26(11):2223-2225. doi: 10.1111/tbj.14025. Epub 2020 Sep 10.PMID: 32909653 Review.

These are now included in the bibliography.

Reviewer 3 Report

Comments and Suggestions for Authors

Dear Authors,

Thank you very much for allowing me to express my opinions related to your work. As a researcher myself, I admire and respect the effort you put into constructing your study and building this manuscript.

Bellow, you can find my comments regarding certain issues. I hope these comments will help you improve both your current and future work.

Title: he title is clear, targeted, and includes relevant keywords

Simple Summary (p.1): it seems you forgot to replace the MDPI template with your summary;

Abstract

abstract says “in this review we discuss…” but does not specify what kind of review this is. Narrative? Systematic?

Introduction

The statement “microsurgical reconstruction remains the most definitive and durable option” seems a bit strong – it should be tuned down a little ore backed up by more references

Patient Evaluation and Risk Stratification - This section reads like a manual. It lists many risk factors (smoking, diabetes, chemo, etc.), but nothing about how much each factor increases risk or which one is most important. No risk scoring system mentioned. Phrases like “we recommend using specific radiologists” are more anecdotal than evidence. Needs references and maybe a table to organize the risks.

Flap Selection and Design - Figure 1 – please use the same font as the rest of the manuscript; also, the font size seems a bit too big; no need for red font color

Alternative Donor Sites & Modifying Flaps - Again, very descriptive. PAP, SGAP, LAP are called “valuable” but there are no statistics to prove how often they succeed or fail. Claims about “less pliable” or “more ideal for volume” are not quantified.

Intraoperative Techniques and Adjuncts - ICG angiography is praised a lot, but no limitations are given (cost, availability, lack of standard cut-offs, false positives).

Recipient Vessel Strategies - Very long and descriptive but again no synthesis.

Revision-Specific Strategies - The section talks about salvage after implant loss or infection, but it’s vague (“may be attempted”, “often preferred”). No real numbers are given about success rates. Statements like “even superficial infections should prompt aggressive interventions” are too general and not nuanced.

Postoperative Management and ERAS - The authors mention ERAS, NIRS, Doppler monitoring etc. but without data. Again, everything is just listed. We don’t know which is actually effective in irradiated/revision patients, and what are the limitations.

Outcomes -  The authors cite “>95% flap success even in high risk” but do not stratify by radiation or revision, nor by flap type. Statements about higher complications in irradiated cases are made but no percentages or odds ratios. BREAST-Q is mentioned but no actual scores.

Conclusions - The final paragraph is vague (“favorable outcomes are consistently achievable”). This is not true for all patients and is not supported by the data presented. The conclusion should instead clearly state what is known (microsurgery is possible, but higher risks), what is unknown (which flap is best in revision cases, role of ICG etc.), and what research is needed (prospective registries, cost-effectiveness studies).

Thank you very much for allowing me to express my opinions.

Sincerly,

Author Response

Manuscript Number: cancers-3895389

Title: Microsurgical Strategies in Post-Radiation and Revision Breast Reconstruction: Optimizing Outcomes in High-Risk Patients

Reviewer #3:

Thank you very much for allowing me to express my opinions related to your work. As a researcher myself, I admire and respect the effort you put into constructing your study and building this manuscript. Bellow, you can find my comments regarding certain issues. I hope these comments will help you improve both your current and future work.

  1. Abstract says “in this review we discuss…” but does not specify what kind of review this is. Narrative? Systematic?

We thank the reviewer for this helpful comment. The manuscript was intended as a narrative review, focused on integrating current literature with institutional experience and expert opinion to guide microsurgical decision-making in complex cases. We have revised to manuscript to clarify (Line 25, Lines 44-45).

  1. The statement “microsurgical reconstruction remains the most definitive and durable option” seems a bit strong – it should be tuned down a little ore backed up by more references

Thank you for this comment. We have now modified the statement to read: “In these radiated and revision scenarios, microsurgical reconstruction may be the most definitive and durable option”

  1. Patient Evaluation and Risk Stratification - This section reads like a manual. It lists many risk factors (smoking, diabetes, chemo, etc.), but nothing about how much each factor increases risk or which one is most important. No risk scoring system mentioned. Phrases like “we recommend using specific radiologists” are more anecdotal than evidence. Needs references and maybe a table to organize the risks.

Thank you for this comment. We have adjusted this section to increase brevity and include more references (Lines 56-60).

  1. Flap Selection and Design - Figure 1 – please use the same font as the rest of the manuscript; also, the font size seems a bit too big; no need for red font color

Thank you for this suggestion. We have made Figure 1 font, font size and font color the same as the rest of the manuscript.

  1. Alternative Donor Sites & Modifying Flaps - Again, very descriptive. PAP, SGAP, LAP are called “valuable” but there are no statistics to prove how often they succeed or fail. Claims about “less pliable” or “more ideal for volume” are not quantified.

We thank the reviewer for these points. We have now modified that section to include more references and quantify claims.

  1. Intraoperative Techniques and Adjuncts - ICG angiography is praised a lot, but no limitations are given (cost, availability, lack of standard cut-offs, false positives).

We thank the reviewer for this excellent point. We agree that areas of conflicting evidence should be clearly acknowledged to present a balanced and transparent synthesis. We have now modified the manuscript to present a more balance assessment.

  1. Recipient Vessel Strategies - Very long and descriptive but again no synthesis.

We thank the reviewer for this point. We have now made modifications to this section as well as included a table (Table 1) to address these comments.

  1. Revision-Specific Strategies - The section talks about salvage after implant loss or infection, but it’s vague (“may be attempted”, “often preferred”). No real numbers are given about success rates. Statements like “even superficial infections should prompt aggressive interventions” are too general and not nuanced.

Thank you for these comments. We have now edited this section to include more detailed discussions on the literature as well as removed generalized statements.

  1. Postoperative Management and ERAS - The authors mention ERAS, NIRS, Doppler monitoring etc. but without data. Again, everything is just listed. We don’t know which is effective in irradiated/revision patients, and what are the limitations.

Thank you for these comments. We have now modified this section to be more succinct and narrative with our personal experiences as well as mentioned comments specifically for radiated patients.

  1. Outcomes - The authors cite “>95% flap success even in high risk” but do not stratify by radiation or revision, nor by flap type. Statements about higher complications in irradiated cases are made but no percentages or odds ratios. BREAST-Q is mentioned but no actual scores.

We appreciate this comment. We have now mentioned specifically success rates stratified by radiated breasts and included references. We have further included BREAST-Q scores for autologous reconstruction as well.

  1. Conclusions - The final paragraph is vague (“favorable outcomes are consistently achievable”). This is not true for all patients and is not supported by the data presented. The conclusion should instead clearly state what is known (microsurgery is possible, but higher risks), what is unknown (which flap is best in revision cases, role of ICG etc.), and what research is needed (prospective registries, cost-effectiveness studies).

We appreciate this thoughtful and important comment. We agree that our prior concluding statement was overly broad and could be misinterpreted as minimizing the risks associated with microsurgical reconstruction in post-radiation and revision settings. We have now modified our conclusion.

Round 2

Reviewer 3 Report

Comments and Suggestions for Authors

Thank you for carefully addressing all my comments and suggestions. I truly appreciate the effort you invested in revising the manuscript. The changes you implemented have notably strengthened the clarity, rigor, and overall impact of your work. I am confident that the paper is now substantially improved and will make a valuable contribution to the field.